

# Morphology of GPS and DPS-TEC Over an Equatorial Station: Validation of IRI and NeQuick 2 Models.

Olumide O. Odeyemi[1] , Jacob Adeniyi[2] , Olushola Oladipo[3], Olayinka Olawepo[3], Isaac Adimula[3], Elijah Oyeyemi[1]

[1] Department of Physics, University of Lagos, Nigeria olumidephysics@yahoo.com, oodeyemi@unilag.edu.ng
[2] Department of Physical Sciences, Landmark University, Omu-Aran, Nigeria adeniyi.jacob@lmu.ng.edu
[3] Physics Department, University of Ilorin, Ilorin

*Correspondence to*: Olumide O. Odeyemi (olumidephysics@yahoo.com)

## 0.0 Abstract

We investigated total electron content (TEC) at Ilorin (8.50$^{o}$N 4.65E, dip lat. 2.95) during a low solar activity 2010. The investigation involved the use of GPS derived TEC, TEC estimated from digisonde portable sounder data (DPS-TEC), the International Reference Ionosphere model (IRI-TEC) and NeQuick 2 model (NeQ-TEC). The five most quietest days of the months obtained from the international quiet days (IQD) from the website http://www.ga.gov.au/oracle/geomag/iqd_form.jsp were used for the investigation. During the sunrise period, we found that the rate of increases in DPS-TEC, IRI-TEC and NeQ-TEC were higher with respect to GPS-TEC. One reason for this can be alluded to an overestimation of plasmaspheric electron content (PEC) contribution in modeled TEC and DPS-TEC. A correction factor around the sunrise where a significant percentage difference of overestimations between the modeled TEC and GPS-TEC was obtained will correct the differences. Our finding revealed that during the daytime when PEC contribution is known to be absent or insignificant, GPS-TEC and DPS-TEC in April, September and December predicts TEC very well. The lowest discrepancies were observed in May, June and July (June solstice) between the observed and all the model values in all hours. There is an overestimation in DPS-TEC that could be due to extrapolation error while integrating from the peak electron density of F2 (NmF2) to around ~ 1000 km in the Ne profile. The underestimation observed in NeQ-TEC must have come from the inadequate representation of contribution from PEC on the topside of NeQ model profile whereas the exaggeration of PEC contribution in IRI-TEC amount to overestimations of GPS-TEC. The excess bite-out observed in DPS-TEC and NeQ-TEC show the indication of overprediction of fountain effect in these models. Therefore, the daytime bite-out observed in these two models require a modifier that could moderate the perceived fountain effect morphology in the models accordingly. Seasonally, we found that all the TECs maximize and minimize during the March equinox and June solstice, respectively. Therefore, GPS-, DPS-, IRI-



and NeQ-TEC reveal the semi-annual variations in TEC as reported in all regions. The daytime
DPS-TEC performs better than the daytime IRI-TEC and NeQ-TEC in all the months, however,
the dusk period requires attention due to highest percentage difference recorded especially for
DPS-TEC and the models in March, and November and December for DPS-TEC.

## 1.0  Introduction

Total electron content (TEC) is the total number of free electrons in a columnar of one
square meter along the radio path from the global positioning system (GPS) satellite to the
receiving station on the Earth. TEC exhibits diurnal, seasonal, solar cycle and geographical
variations. Therefore, the physical and dynamical morphology of the TEC over a given location
is of great importance in trans-ionospheric communications during undisturbed and disturbed
geomagnetic conditions (Jesus et al., 2016; Tariku, 2015; and Akala et al., 2012). GPS-TEC is
quantified from the GPS orbiting satellites to the GPS receiver station on the Earth, with an
approximate distance of 20200 km (Liu et al., 1996b; Rama Rao et al., 2006a; Rama Rao et al.,
2006b; Liu et al., 2006). Thus, a typical GPS-TEC measurement includes the complete
plasmaspheric electron content (PEC).

The International Reference Ionosphere (IRI) is a standard model that is based on
worldwide data from various measurements (Bilitza, 2001; Bilitza, 1999; Bilitza, 1986; Bilitza
and Rawer, 1998; and Rawer et al., 1978 ). The Committee on Space Research (COSPAR) and
the Union Radio-Scientifique Internationale (URSI) meet yearly to improve the IRI model. The
IRI model provides reliable ionospheric densities, composition, temperatures, and composition
(Bilitza, 2001). The Comite Consultatif International des Radiocommunications (CCIR) Model
was developed by Rawer and Bilitza (1990) while the Union Radio Scientific International
(URSI) developed URSI option of IRI model (Fox and McNamara, 1988; Rush et al., 1989). The
latest version of IRI model can be found at all time on the web
(http:nssdc.gsfc.nasa.gov/space/model/ions/iri.html) with improvements on earlier versions of
the model from the working group scientists on the model. The International Telecommunication
Union, Radio-communication Sector (ITU-R) has introduced and adopted NeQuick for TEC
modeling.  In the NeQuick 2, the position, time and solar flux or sunspot number over a given
location are embedded in the NeQuick model code. The output of the NeQuick 2 program is the



electron density along any ray-path while the corresponding TEC measurement is by numerical
integration in space and time. The availability ionospheric parameters as contribution for global
ionospheric models are not sufficient over the Africa sector compared to the consistent input of
the parameters from the Asia and America sectors.  Therefore, the investigations of the
ionospheric parameters over Africa are continuously required to improve the global ionospheric
model. For example, Bagiya et al. (2009) studied TEC around equatorial-low latitude region at
Rajkot (22.29° N, 70.74° E, dip 14.03° N )  during low solar activity and showed that TEC
revealed seasonal variation with maximum and minimum at March equinox and June solstice,
respectively.  Young et al. (1970) examined a night enhancement in TEC at equatorial station of
Hawaii and reported that the nighttime enhancement in TEC showed seasonal and solar cycle
dependences.   Olwendo et al. (2012) investigated TEC in Kenyan and found a semi-annual
variation with minimum and maximum TEC June solstice and March equinox, respectively.
They further reported that the TEC had a noontime dip and day-to-day variability.  Using
Faraday rotational technique, Olatunji (1967) studied TEC variation over equatorial station at
Ibadan and found no daytime bite-out and seasonal anomaly over the equatorial region.  Adewale
et al. (2012) investigated TEC at Uganda during low and high solar activities. They found that
TEC was higher during high solar activity compared with the low solar activity. Karia and
Pathak. (2011) investigated the TEC at Surat and  showed that TEC maximizes and minimizes
during the equinox and June solstice, respectively.  Rastogi et al. (1975) investigated the diurnal
variation of TEC using Faraday rotation over the magnetic equator.  They found that the TEC at
the topside was higher than the TEC at the bottomside during the nighttime, however during the
daytime, equal distribution of TEC was found on the topside and the bottomside of electron
density (Ne) profile.

The DPS-TEC is the combination of  TEC from the bottomside and topside electron
density (Ne). The topside DPS-TEC  is an extrapolated TEC from the peak electron density of
the F2 region (NmF2) to around~ 1000 km (DPS-TEC) thus, the major PEC contribution from
the greater altitudes is excluded from DPS-TEC measurement (Belehaki et al., 2004; Breed and
Goodwin, 1997; and Reinisch and Huang, 2001). The combined investigation on GPS  and DPS
is scanty over Africa (Ciraolo and Spalla, 1997) due to lack of colocated GPS and DPS data in
most of the equatorial stations. Therefore, the ionospheric modeling and the improvement on the



existing models are important to understanding of the ionospheric structure of a given location in
the absence of instrumentations.
Regarding the DPS-TEC measurement, Barbas et al. (2010) investigated GPS-TEC and
DPS-TEC at Tucuman (26.69°S, 65.23°W) during different seasons and magnetic activities.
They concluded that the DPS-TEC variation represented the GPS-TEC in all hour with minimal
discrepancy. Reinisch et al.(2004) investigated GPS-TEC from satellite beacon signals and
DPS-TEC from the DPS at mid-latitude and equatorial region. They found that GPS-TEC and
DPS-TEC variations were similar. However, the daytime GPS-TEC profile values were higher
than DPS-TEC profile values. Belehaki et al. (2004) extracted the plasmaspheric electron
content (PEC) from the GPS-TEC at Athens (38°N, 23.5°E) over a year and found a maximum
and minimum contribution of PEC in the morning and evening, respectively. Zhang et al. (2004)
investigated the simultaneous variation of DPS-TEC and GPS-TEC over Hainan and reported
that the daytime DPS-TEC and GPS-TEC variations are close, however during nighttime to pre-
sunrise, a significant discrepancy between DPS-TEC and GPS-TEC was observed. Mosert and
Altadill (2007), Jodogne et al. (2004) and Mckinnell et al. (1996) concluded that estimated
PEC from the GPS-TEC and DPS-TEC is possible in colocated GPS and DPS station.

Rios et al. (2007) investigated the DPS-TEC and IRI-TEC and found a smaller DPS-TEC
compared to IRI TEC in all hour. McNamara (1985) observed discrepancies between DPS-TEC
and IRI-TEC, he found that during the daytime, the IRI underestimated the observed DPS-TEC.
Obrou et al. (2008) compared the DPS-TEC and IRI-TEC at Korhogo during high and low solar
activity. They found that DPS-TEC and IRI-TEC values were close during high solar activity
(HSA) and low solar activity (LSA). Nevertheless, the performance between the observed and
model TEC was better in HSA compared to LSA. Adewale et al. (2012), Okoh et al. (2014), Jee
and Scherliess (2005), Sulungu et al. (2017), and Migoya Orué et al. (2008) validated the IRI-
TEC with GPS-TEC in different regions and found high discrepancies between the IRI and
observed TEC. Furthermore, Arunpold et al. (2014) and Olwendo et al. (2012; 2013) also
concluded that the signature of the geomagnetic storm was absent in the morphology of IRI.
Thus IRI-TEC could not predict the effect of the geomagnetic storm on observed TEC.



Regarding the studies on NeQuick model, Cherniak and Zakharenkova (2016) validated NeQuick model and found that the topside ionosphere above ~ 500km in the NeQuick model consistently revealed underestimation due to inaccurate representation of topside Ne profile. Bidaine and Warnant (2011) validated NeQuick model with slant total electron content (STEC). Rabiu et al. (2014) validated NeQuick model using the seasonal variation of TEC over equatorial station of Africa. They found that the upper boundary of the NeQuick models up to 20,000 km needs to be adjusted to accommodate the PEC-TEC in NeQuick model. Migoya-Orue et al. (2017) introduced B2bot in NeQuick and reported the improvement in the topside performance of the NeQuick model in the computation of TEC. Andreeva and Lokota (2013) found that the NeQuick reproduced the maximum values of electron density observed in the experiments. However, the electron density profiles reproduction from NeQuick show significant discrepancies in some periods. Leong et al. (2013) investigated TEC and NeQuick 2 models. They found that the observed TEC and NeQuick 2 TEC are close in values during post-noon and post-midnight. However, the post-sunset revealed some discrepancies. Yu et al. (2012) investigated the monthly average of NeQuick model over three stations in China (Changchun, Beijing, and Chongqing) during the quietest period. They found that NeQuick accurately predicted GPS-TEC. However, the NeQuick underestimated the observed TEC and NmF2 in few cases.

The current contributions of Africa on the improvement of ionospheric models (IRI and NeQuick) are not adequate compared with the continuous support received from Asia and South America. The scanty of ionospheric instrumentations at the equatorial region of Africa has a considerable effect on the shortcoming. Therefore, the continuous validation of IRI and NeQuick models with the observed parameter is necessary for improved ionospheric model. Furthermore, the investigation on DPS-TEC has not been reported extensively for comparison purpose. Therefore, this paper set to investigate the combined relationship between the variations of GPS-TEC and DPS-TEC, and validations of IRI-TEC, and NeQ-TEC models with the observed parameters. Our finding will reveal the suitability of DPS-TEC, IRI-TEC and NeQ-TEC in place of GPS-TEC. The result will also reveal the appropriate model for the equatorial station in Africa. Thus, the changed TEC obtained from the combined relationship between GPS-TEC,



DPS-TEC, IRI-TEC and NeQ-TEC could be used to improve the discrepancy in the model
values.

## 2.0  Methods of Analysis of GPS and DPS Data

The five most quiet days of GPS and DPS-TEC data for each month were presented and

analyzed during the year 2010 with the local time (LT).

### 2.1  GPS-TEC

The Slant TEC records from GPS has errors due to satellite differential delay  (satellite

bias (bs)) and receiver differential delay (receiver bias (br)) and receiver inter-channel bias ($b_{SR}$).
This uncorrected slant GPS-TEC measured at every one-minute interval from the GPS receiver
derived from all the visible satellites at the Ilorin station are converted to vertical GPS-TEC
using the relation below in equation (1).
$(GPS - TEC)_V = (GPS - TEC)_S - [b_S + b_R + b_{SR}]/S(E)$                    1
Where $(GPS - TEC)_S$ is the uncorrected slant GPS-TEC measured by the receiver, $S(E)$ is the
obliquity factor  with zenith angle (z) at the Ionospheric Pierce Point (IPP), E is the elevation
angle of the satellites in degrees and $(GPS - TEC)_V$  is the vertical GPS-TEC at the IPP. The
$S(E)$ is given as

$$S(E) = \frac{1}{\cos(z)} = \left[1 - \left(\frac{R_E \times \cos(z)}{R_E + h_s}\right)^2\right]^{-1/2}$$                    2
Where $R_E$ is the mean radius of the Earth measured in kilometer (km), and $h_S$ is the height of the
ionosphere from the surface of the Earth, which is approximately equal to 400 km according to
Langley et al. (2002) and Mannucci et al. (1993). The ten most quiet slant GPS-TEC data for
each month in the year 2010 were analyzed using Krishna software. This software reads raw data
and corrects all source of errors mentioned above from Global Navigation Satellite System
service (IGS) code file. A minimum elevation angle of 20 degrees is used to avoid multipath
errors. The estimated vertical GPS-TEC data is subjected to a two sigma (2σ) iteration. This
sigma is a measure of GPS point positioning accuracy. The average one-minute VTEC data were
converted to hourly averages.



## 2.2 DPS-TEC

Regarding the total electron content (TEC) from the digisonde portable sounder (DPS), the Standard Archive Output (SAO) files obtained from the recorded of ionogram from the installed DPS at the University Ilorin were edited to remove magnetically disturbed days. Huang and Reinisch (2001) technique was used to compute the DPS-TEC. The complete vertical DPS-TEC computation is obtained by applying the integration over the vertical electron density (Ne(h)) profile as shown in the equation below.

$$\text{TEC} = \int_0^{\text{hmF 2}} \text{Ne}_B (dh) + \int_{\text{hmF 2}}^{\infty} \text{Ne}_T (dh) \qquad 3$$

Where $\text{Ne}_B$ and $\text{Ne}_T$ are the bottomside and topside Ne profiles, respectively. The $\text{Ne}_B$ is computed from the recorded ionograms by using the inversion technique developed by Huang and Reinisch (1996). It is known that the information above the peak of the F2 layer is absent from the record of the ionogram. Thus the $\text{Ne}_T$ is computed by approximating the exponential functions with suitable scale height (Bent et al., 1972) with less estimated error of 5%. The ionograms are manually scaled and inverted into electron density profile using the NHPC software and later processed with the SAO explorer software based on the technique described above to obtain the TEC (Reinisch et al., 2005). An average of TEC for each hour is computed over the selected days. The universal time (UT) is the time convention for these analyses (GPS and DPS data). Local time (LT) was used in this study. Thus, 0100 UT (Universal time) is the same as 0200 LT (Local Time) in Nigeria. In this study, the seasonal variation was arranged into four seasons, as, March equinox or MEQU (March, and April), June solstice or JSOL (June, and July), SEQU (September, and October) and December solstice or DSOL (November, December). Due to technical reasons, there were data gaps in all days during January and February in the DPS measurements, therefore, we decided to neglect data in January and February in GPS-, IRI-, and NeQuick measurements for comparison purposes thus, two simultaneous representative months were used to infer each season. The average of the monthly median of the five quietest days for the representative months is found to give each parameter a particular season discussed above.

## 2.3 Validation of IRI - 2016 and NeQuick Models

The observed TEC and NmF2 were compared with the IRI-2016 model. The website http://www.ccmc.gssfc.nasa.gov/modelweb/models/iri_vitmo.php provides the modeled TEC



values. The upper boundary height 2000 km was used, and the B0 table option was selected for
the bottomside shape parameter. The equations 3a, 3b and 3c represent the difference between
GPS-TEC and DPS-TEC, GPS-TEC and IRI-TEC and GPS-TEC and NeQ-TEC while equations
4a, 4b, and 4c below show the percentage change between GPS-TEC and DPS-TEC, GPS-TEC
and IRI-TEC, and GPS-TEC and NeQ-TEC.

$\Delta_{GPS/DPS} = DPS_{TEC} - GPS_{TEC}$            3a
$\Delta_{GPS/IRI} = IRI_{TEC} - GPS_{TEC}$            3b
$\Delta_{GPS/NeQ} = NeQ_{TEC} - GPS_{TEC}$            3c
$\%(\Delta_{GPS/DPS}) = \frac{DPS_{TEC} - GPS_{TEC}}{DPS_{TEC}} \times 100$            4a
$\%(\Delta_{GPS/IRI}) = \frac{IRI_{TEC} - GPS_{TEC}}{IRI_{TEC}} \times 100$            4b
$\%(\Delta_{GPS/NeQ}) = \frac{NeQ_{TEC} - GPS_{TEC}}{NeQ_{TEC}} \times 100$            4c
$\Delta_{GPS/DPS}$, represents the change between GPSTEC and DPS-TEC
$\Delta_{GPS/IRI}$, represents the change between GPS-TEC and IRI-TEC
$\Delta_{GPS/NeQ}$ represents the change between GPS-TEC and NeQ-TEC
$\%(\Delta_{GPS/DPS})$, represents the percentage deviation between GPS-TEC, and DPS-TEC
respectively.
$\%(\Delta_{GPS/IRI})$, represents the percentage deviation between GPS-TEC, and IRI-TEC respectively.
$\%(\Delta_{GPS/NeQ})$, represents the percentage deviation between GPS-TEC, and NeQ-TEC
respectively.

The Abdus Salam International Centre for Theoretical Physics (ICTP) - Trieste, Italy
with the collaboration of the Institute for Geophysics, Astrophysics and Meteorology (IGAM) of
the University of Graz, Austria developed the web front-end of NeQuick. This quick-run
ionospheric electron density model developed at the Aeronomy and Radiopropagation
Laboratory modeled TEC along any ground-to-satellite straight line ray-path. Therefore, the
observed TEC use for the validation of the NeQuick 2 was obtained in the address below
https://t-ict4d.ictp.it/nequick2/nequick-2-web-model.

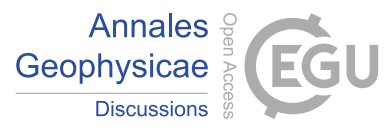

## 3.0 Result

### 3.1 Monthly Median Variations

Figure 1 shows the plots of diurnal variations of the monthly median of GPS-, DPS-, IRI-, and NeQ- TEC during quiet period. The GPS-TEC is plotted in black line with the star symbol; the DPS-TEC is in green with the diamond symbol, IRI-TEC is in red line with zero symbols, and finally, the NeQ-TEC is in blue line with multiplication symbol. All TEC plots are regulated by the same local time (LT) on the horizontal axis. We found that the variations of GPS-, DPS-, IRI, and NeQ-TEC increase gradually from the sunrise period and reach the daytime maximum, then later decay till it gets to a minimum around 0500 or 0600 LT. These results show that the models capture the well known solar zenith angle dependence of TEC. Regarding the GPS-TEC, the pre-sunrise minimum is ranged between ~0.43 TECU (June) to ~2.35 TECU (April ) and the sunrise minimum of ~ 1.76 TECU, ~ 2.58 TECU, and ~ 2.58 TECU are observed in March, November, and December respectively. The daytime maximum ranged between ~ 20 TECU (June) - ~ 35.4 TECU (November) and occurred around 1500 - 1700 LT. The dusk time decay in GPS-TEC is faster in June and slower in November around 2400 LT. Regarding DPS-TEC, the pre-sunrise minimum of DPS-TEC ranged between ~ 0.66 TECU (August) - ~ 4.59 TECU (May) around 0500 LT, while the daytime maximum is found around 1000 - 1600 LT and ranged between ~ 24.2 TECU (July) - ~ 38.0 TECU (March). A moderate daytime bite-out in DPS-TEC was observed in March, May, August, September, October, November and December. The duration of the bite-out was longer in October (1000 -1600 LT). The decay of DPS-TEC is faster in June and lower in April. Regarding the IRI-TEC, the pre-sunrise minimum in IRI-TEC ranged between ~ 2.3 TECU (March) - ~ 4.1 TECU (October and November) and found around 0500 LT. The daytime maximum is seen around 1500 LT and 1600 LT and ranged between ~ 21.9 TECU (July) - ~31.7 TECU (November). A moderate bite-out is present in all  months between 1100 - 1600LT. As regard NeQ-TEC, the pre-sunrise minimum ranged between ~ 1.31 TECU (July) - ~ 2.88 TECU (December) and found around 0500 LT. The daytime maximum around 1000 LT and 1600 LT, ranged between ~ 17.75 TECU (July) -  ~ 25.45 TECU (November). A moderate noon time bite-out is seen in May, June, July, and August within a short time range.  The decay in NeQ-TEC is faster in July and slower in November.

Our investigation reveals that GPS, DPS, IRI and NeQ-TEC  decay is faster and slower in June and December seasons, respectively. The maximum daytime is found in the DPS-TEC, whereas





the minimum daytime is observed in NeQ-TEC.  The DPS-TEC show a higher pre-sunrise
minimum of ~ 4.59 TECU (May) while the GPS-TEC revealed a smaller pre-sunrise minimum
of ~ 0.43 TECU (June).

**3.2  Percentage deviation of DPS-TEC**

Figures 2a, and 2b show hourly variations of the changed TEC, and mass plot of hourly

variations of % changed TEC between  GPS-TEC and DPS-TEC from March to December
during quiet period. Between 0100 - 0500 LT (Figure 2a),  DPS-TEC constantly lower than the
GPS-TEC in March, April, August, September, November, and December except in June and
July and the changes range between ~ -4.67 TECU (November) - ~ - 0.53 TECU (August). In
Figure 2b,  DPS-TEC uniformly overestimated GPS-TEC around 0700 - 1500 LT in all month
except in June (0700 LT), November (1500 LT), August and December (1400 - 1500 LT) where
DPS-TEC underestimated GPS-TEC.   The percentage of overestimation  ranges between ~ 2%
(November) - ~ 49% (March). We also observed that the DPS-TEC underestimated GPS-TEC
between 1700 - 2400 LT in all months and ranged between  ~ - 0.15% (October) - ~ - 306%
(November). A few cases of overestimation are noticed in March, May and September around
local time (1700 LT).  We also notice a consistent overestimation of DPS-TEC around 0100 LT
and 0400 LT in June and July while underestimation occurred in March, April, August,
September, October, November and December within the same period.

**3.2  Percentage deviation of IRI-TEC**

Figures 3a, and 3b, give hourly variations of the changed TEC, and mass plots of hourly

variations of % changed TEC  between  GPS-TEC and IRI-TEC from  March to December. The
change  between  IRI-TEC and GPS-TEC occurred between 0100 - 1200 LT in all months except
in March and April and the changed TEC ranged between ~ 0.01 TECU (November)  to ~ 15
TECU (October). The IRI-TEC continually overestimated GPS-TEC around 0100 - 1200 LT in
all months however, underestimation occurred  in March (0100 - 0500 LT) ,  April  (1200 LT),
September and November (0300 - 0400 LT) . The overestimation percentage ranges between ~
0.1% (December) - ~ 86% (June) between 0100 - 1200 LT. We also observed that in May and
June,  IRI-TEC overestimated GPS-TEC during May and June in all hours  between ~ 2% (1900
LT)  and  ~ 86% (0500 LT), respectively. Between 1300 - 2400 LT, we observed some irregular



patterns of underestimation and overestimation of DPS-TEC over GPS-TEC in most of the months.

### 3.4 Percentage deviation of NeQ-TEC

Figures 4a, and 4b, reveal the hourly variations of the changed TEC, and mass plots of hourly variation of % changed TEC between GPS-TEC and NeQ-TEC from March to December during quiet period. The increase change between NeQ-TEC and GPS-TEC are found 0100 - 0900 LT except in November and December. We also found that, NeQ-TEC constantly overestimated GPS-TEC around 0100 - 0900 LT in all month except in March, April, August, September and November around 0400 LT and also around 0500 LT in March, April and November. The overestimation percentage is ranged between ~ 0.02% (April) - ~ 81% (July). We also observed that the NeQ-TEC underestimated GPS-TEC between 1200 - 1900 LT in all months and ranged between ~ - 0.3% (October) - ~ - 75% (November). In July, we noticed a consistent overestimation of NeQ-TEC in all hours except around 1300 LT (~ - 1.5% ) and 1400 LT (~ - 0.6%) . Between 2000 - 2400 LT, NeQ-TEC overestimated GPS-TEC in March, April, June, September, October and December whereas in May, July, August and November, NeQ-TEC underestimated GPS-TEC.

### 3.5 Comparisons of the deviations from GPS-TEC

From Figure 2b, 3b, and 4b, we constantly found high percentage of underestimations of DPS-TEC, IRI-TEC and NeQ-TEC with respect to GPS-TEC between 0400 - 0600 LT in March and December. Around 0100 - 0500 LT, highest DPS-TEC percentage of underestimation are ~ 190%, ~ 210% - ~ 280% in March, November, and December respectively, highest IRI-TEC percentage of underestimation of IRI-TEC is ~ 200% in March, and highest NeQ-TEC overestimations is ~ 68% and ~75% in June and July, respectively and highest underestimation is ~ 60% in March. Between 0700 LT - 1800 LT, DPS-TEC overestimation and underestimation ranges between ~ 10% - ~ 10% in all months, IRI-TEC overestimation and underestimation ranges between ~ 70 - ~ 50% in all months, and NeQ-TEC overestimation and underestimation of ranges between ~ 80% - ~ 80% in all months. During 1900 - 2400 LT, DPS-TEC highest underestimation is ~ 310% in March, IRI-TEC overestimation and underestimation are found between the range of ~ 50% - ~ 50% in all months, and NeQ-TEC overestimation and



underestimation ranges between ~ 70% - ~ 70% in all months. Figure 5 reveals the seasonal
variations of GPS-TEC, DPS-TEC, IRI-TEC, and NeQ-TEC during quiet period. We observed
that both DPS-TEC and models reproduce the semi-annual variation with maximum and
minimum TEC at March equinox and  June solstice, respectively. The daytime maximum is
ranged between  ~ 24.8 TECU  (NeQ) - ~ 34  TECU  (DPS), ~ 19.2 TECU (NeQ) - ~ 22.6 TECU
(DPS), ~ 24.9 TECU (NeQ) - ~ 33.5 TECU (DPS) and ~ 24.55 TECU (NeQ) - ~ 31 TECU
(DPS), in March equinox, June solstice, September equinox, and December solstice,
respectively.

**4.0  Discussion of Result**
An investigation into the variations of GPS-TEC, DPS-TEC, IRI-TEC, and NeQ-TEC at
an equatorial station ($8.5^0$N $4.65^0$ E) in Africa during low solar activity in the year 2010 has been
carried out. The TEC increases gradually from the sunrise period, then slowly reached the
daytime maximum, and later decay till the pre-sunrise minimum. This result indicates that the
TEC is a solar zenith angle dependence revealing maximum and minimum in TEC during the
noontime and pre-sunrise or sunrise minimum, respectively (Wu et al.2008; Aravindan and Iyer
1990; and Kumar and Singh 2009). Interestingly,  the faster increase in the DPS-TEC than GPS-
TEC during pre-sunrise is not consistent with the findings of Ezquer et al. (1992) at Tucumán
(26.9° S; 65.4° W), Belehaki et al. (2004) at Athens in the middle latitude,  McNamara (1985) at
low latitude and Obrou et al. (2008) at Korhogo (9.33°N, 5.43°W, Dip = 0.67°S) and found
smaller  DPS-TEC compared with GPS-TEC.  The evidence of  PEC on GPS-TEC was recently
reported by Belehaki et al. (2004). They extracted the plasmaspheric electron content (PEC)
from the GPS-TEC and found a significant PEC in the morning and evening. Also, Jodogne et al.
(2004), Mosert and Altadill (2007), and Mckinnell et al. (1996)  obtained a rough estimation of
PEC from the GPS and DPS-TEC. They concluded that the combined GPS-TEC and DPS-TEC
could give the PEC of a given location. Therefore, a larger DPS-TEC during the sunrise could be
attributed to inaccurate representation of PEC in the topside DPS-TEC profile during
extrapolation from the peak of NmF2 to around  ~ 1000 km of Ne profile  . Thus, a typical GPS-
TEC naturally includes the PEC measurement ( Belehaki et al. 2003; Balan and Iyer, 1983;
Carlson, 1996; and Breed and Goodwin, 1997).





Furthermore, our observation in GPS-TEC shows no conspicuous noontime bite-out. The
bite-out is attributed to the occurrence of the most active fountain effect during the noontime at
the magnetic equator due to the lifting of ionospheric plasma. Thus, the bite-out result from the
interaction of eastward electric field and earth horizontal magnetic field.  The interactions
resulted to the lifting of plasma at the magnetic equator and diffused along geomagnetic field
lines into the high latitudes, so leaving the reduced TEC at the magnetic equator
(Bandyopadhyay, 1970;  Olwendo et al., 2012; Skinner et al., 1966;  Bolaji et al.,  2012).
However, the absence of daytime bite-out (Olatunji, 1967)  in GPS-TEC found in our result
shows that the productions of the bottomside and topside electron content are enhanced quickly
to replenish the loss of the ionization that occurs during the noontime through the fountain effect.
The  higher DPS-TEC compared with IRI-TEC around sunrise is not consistent with Rios et al.
(2007) who investigated comparison of DPS-TEC and IRI-TEC and found that DPS-TEC is
smaller than IRI TEC in all hour. They concluded that the prediction of IRI-TEC included the
high topside Ne profile. Thus, our observation suggests that the IRI-TEC has included low
topside Ne profile in the model or excessive exaggeration of PEC in the topside Ne profile in the
DPS-TEC.   Our investigation shows that the daytime GPS-TEC and DPS-TEC in April, August
and December appear to be approximately equal. This finding suggests that the topside Ne
profile in DPS-TEC are moderately captured in the topside Ne profile in GPS-TEC. This finding,
thus indicates the absence of PEC profile in DPS-TEC approximately reproduced the daytime
GPS-TEC and IRI-TEC ( April, August and December). The insignificance of daytime PEC in
the observation is inferred from the report of Rastogi et al. (1975) who measured TEC from
Faraday rotations from ground receiver to ~ 20200 km. They found that the PEC contribution on
the topside and the bottomside Ne profile is insignificant during the daytime. Moreover,
Belehaki et al. (2004) investigation has recently reported the negligible PEC contributions during
the daytime.

Our higher daytime DPS-TEC compared with daytime IRI-TEC is consistent with
McNamara (1985) who reported higher DPSTEC compared with IRI-TEC during the daytime.
However, in the report of Obrou et al. (2008) at the equatorial station, the IRI-TEC was higher
than the DPS-TEC at the low solar activity.  We found a reduced daytime IRI and NeQ-TEC
compared with GPS-TEC that  indicates the excessive PEC removal from the model values that



its PEC contribution had been initially exaggerated during the sunrise. Our finding is supported
by  Migoya-Orue et al. (2017), Zakharenkova (2016), Rabiu et al., (2014),  Nava and Radicella
(2009), and Zh et al. (2014). They concluded that the topside ionosphere in the NeQuick model
consistently revealed underestimation of observed TEC.  The daytime IRI-TEC (April, July,
August, and September ) and  NeQ-TEC (June)  is approximately reproduced in the GPS-TEC;
this implies that the  model factors in IRI and NeQ perform best in the absence of significant
PEC contribution.

The hourly variations of percentage difference  between GPS-TEC and DPS-TEC, GPS-

TEC and IRI-TEC and GPS-TEC and NeQ-TEC in all months revealed that the pre-sunrise
values in DPS-TEC, IRI-TEC and NeQ-TEC require an attention due to high percentage
difference recorded in all variations especially in March for DPS-TEC and the models, and
November and December for DPS-TEC.  The daytime DPS-TEC value is closer to the GPS-TEC
value compared to the daytime IRI-TEC and NeQ-TEC values. The nighttime NeQ-TEC and
IRI-TEC perform better  with GPS-TEC compared with DPS-TEC in all months, however more
improvement is also required to minimize the effect of the discrepancies observed during the
night. More work needs to be done during the pre-sunrise in all models especially in March for
all models, and November and December for DPS-TEC.

Seasonally, we discovered that TEC is maximum and minimum during the equinoxes and

the solstices, respectively. Our report is consistent with Mala et al. (2009), Wu et al. (2008),
Kumar and Singh (2009), and Balan and Rao, (1984) who investigated TEC in various regions.
They concluded that the seasonal variation in TEC is attributed to the seasonal differences in
thermospheric composition. Moreover, the sub-solar point is around the equator during the
equinox. Consequently, the sun shines directly over the equatorial region, and in addition to the
high ratio of O/N2 around the region, this translates to stronger ionization, thus, semi-annual
pattern is formed. Our finding is supported by Ross  Skinner et al. (1966),  Bolaji et al.  (2012),
and Scherliess and Fejer (1999) who obtained semi-annual variation in TEC. Scherliess and Fejer
(1999) suggested that daytime E × B drift velocities result to semi-annual variation because the
drift velocities are  more and  less  significant  in  the  equinoctial months  and  June solstice,
respectively.




## 5.0  Conclusion


An investigation into the quietest GPS-TEC, DPS-TEC, IRI-TEC, and NeQ-TEC over an
equatorial station of Africa during just ascending phase cycle of low solar activity in the year
2010 was carried out. Our findings indicate that the variations in GPS, DPS, IRI, and NeQ-TEC
are solar zenith angle dependence having maximum and minimum TEC during the noontime and
pre-sunrise or sunrise minimum. We also found that the absence of daytime bite-out in the GPS-
TEC is exaggerated in the DPS-TEC, IRI-TEC, and NeQ-TEC morphologies. Furthermore, our
result reveals a faster sunrise increase in DPS-TEC, IRI-TEC, and NeQ-TEC than GPS-TEC that
is attributed to the misinterpretation of the topside Ne profile of the DPS-TEC, IRI-TEC, and
NeQ-TEC in order to incorporate the plasmaspheric electron content (PEC) into the models. The
daytime DPS-TEC is also higher than the daytime GPS-TEC, IRI-TEC, and NeQ-TEC, except in
April, September and December where daytime DPS-TEC and GPS-TEC values are close. The
daytime GPS-TEC is also approximately equal the daytime IRI-TEC in April, July, August and
September whereas in the daytime NeQ-TEC only June approximately close to the daytime GPS-
TEC. The close values in daytime TEC obtained in DPS-TEC and IRI-TEC in some months may
be unconnected to the improved model values in the absence or a little PEC contributions during
the daytime. Another finding is the faster decay in DPS-TEC during the dusk time compared to
GPS-TEC, IRI-TEC, and NeQ-TEC. However, the decline is approximately similar in value
found in June, July and August (June solstice).   The hourly variations of percentage difference
between GPS-TEC and DPS-TEC, GPS-TEC and IRI-TEC and GPS-TEC and NeQ-TEC in all
months revealed that the pre-sunrise values in DPS-TEC, IRI-TEC and NeQ-TEC require an
attention. The daytime DPS-TEC value is closer to the GPS-TEC value compared to the daytime
IRI-TEC and NeQ-TEC values. The nighttime NeQ-TEC and IRI-TEC perform better  with
GPS-TEC compared with DPS-TEC in all months.  This study was carried out during the
quietest period of the year 2010; it will be of advantage to investigate the similar work during the
most disturbed days and compared with our results. Moreover, additional stations in the
equatorial region will be needed to validate the latitudinal effect of the model with the observed
parameters. This will reshape the model parameters for improved ionospheric modeling over
Africa.



**6.0 Acknowledgments**
We would like to appreciate Massachusetts University and Boston College, respectively
for the donations of DPS and GPS facilities to the University of Ilorin, Nigeria. Also, the kind
supports of the University of Ilorin in running the DPS and GPS stations and in providing the
data for use are gratefully acknowledged and appreciated.

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

## 660   7.0 Figures





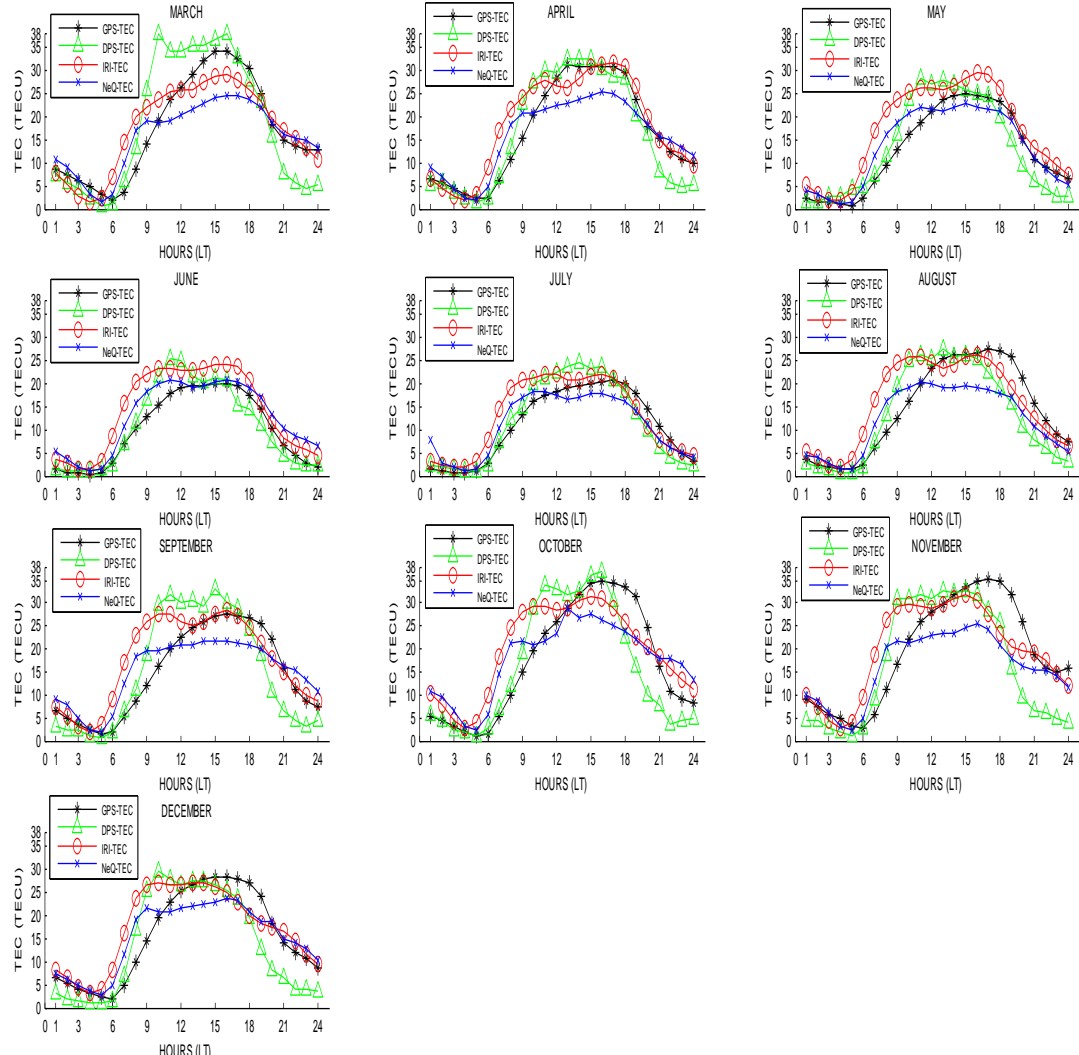

Figure1. The hourly variations of the monthly median of GPS, DPS, IRI, and NeQuick TEC in March-December during quiet period.





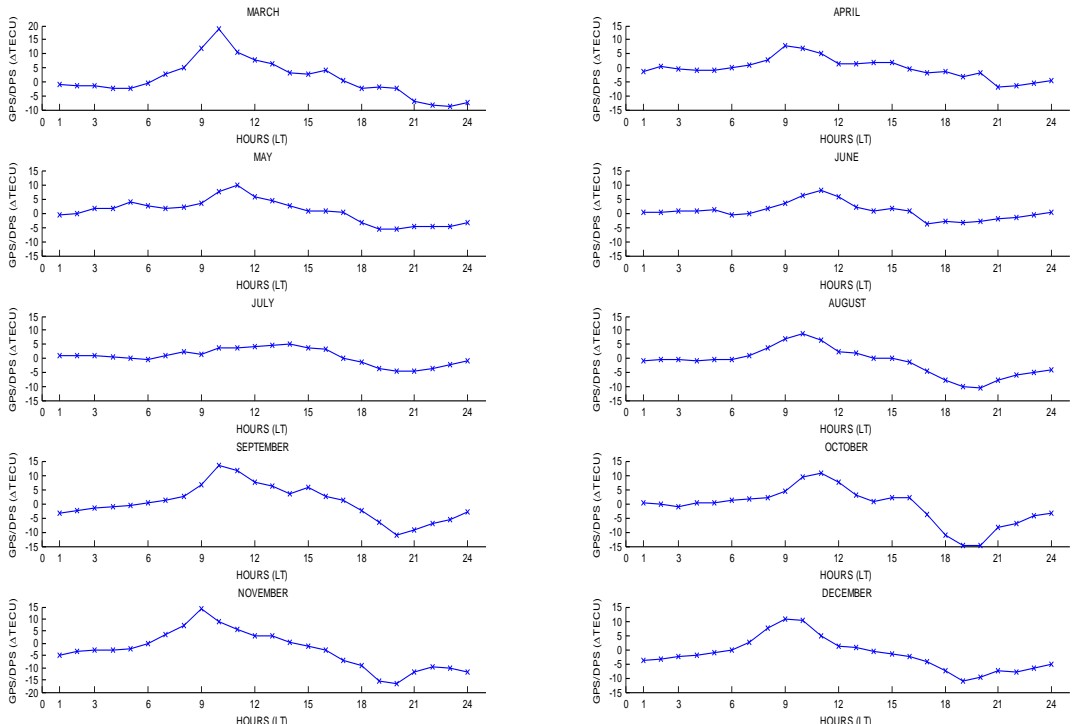

Figure 2a. The hourly ΔTEC variations between the GPS-TEC and DPS-TEC from March -
December during quiet period.

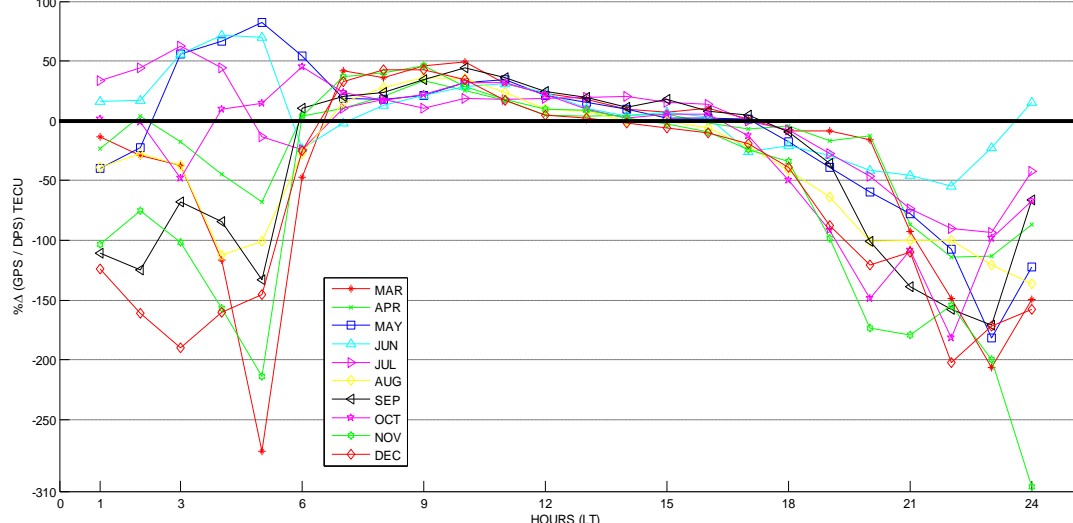

Figure 2b. The mass plot of the hourly %Δ TEC variations between the GPS-TEC and DPS-TEC
from March - December during quiet period.





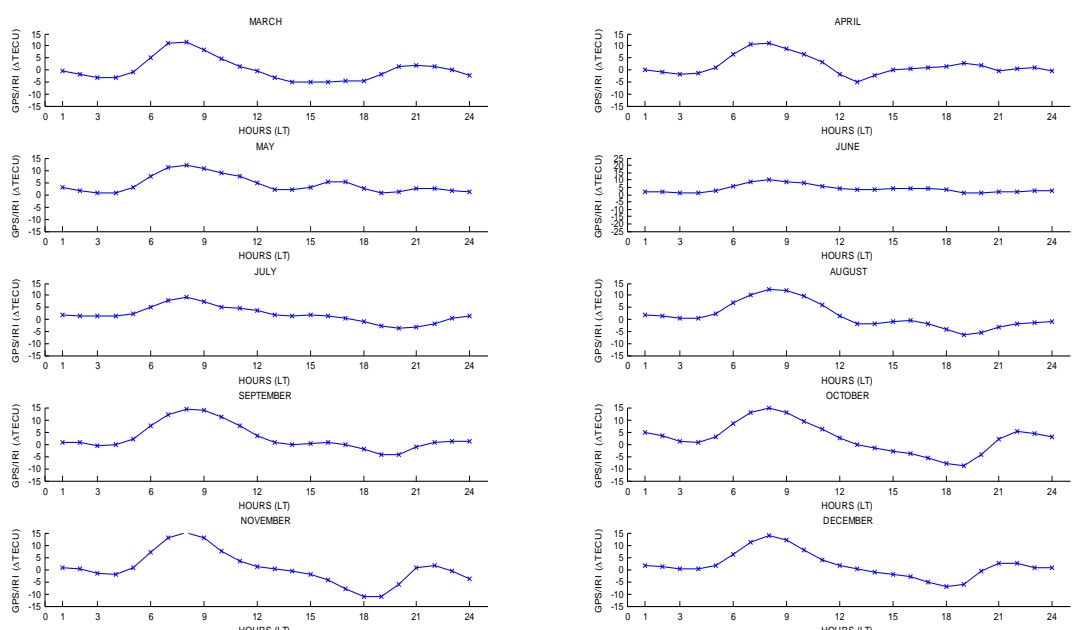

Figure 3a. The hourly ΔTEC variations between the GPS-TEC and IRI-TEC from March -
December during quiet period.

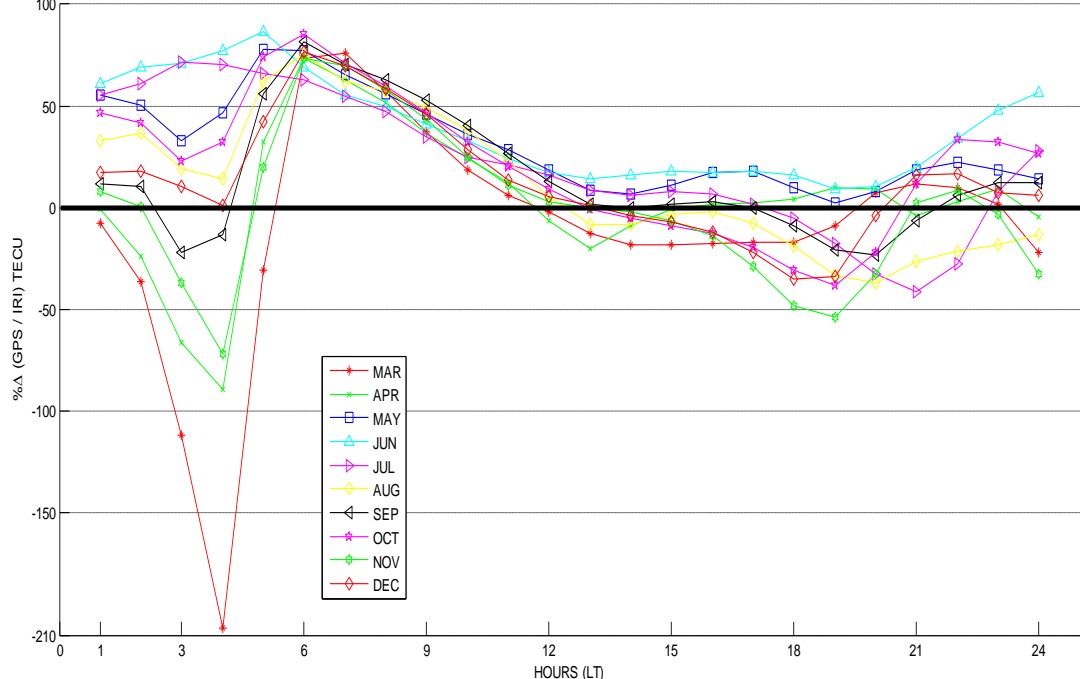

Figure 3b. The mass plot of the hourly %Δ TEC variations between the GPS-TEC and IRI-TEC
from March to December during quiet period.




680   .

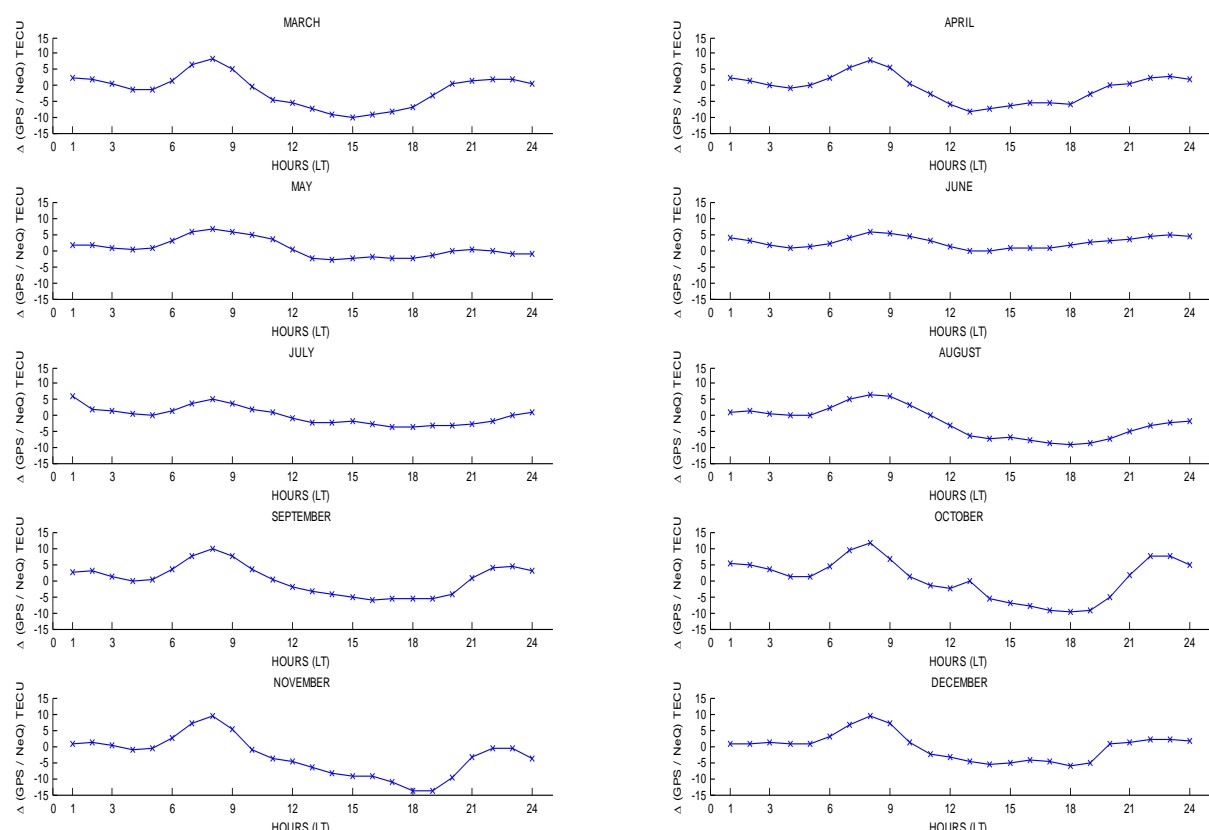

Figure 4a.  The hourly ΔTEC variations between the  GPS-TEC and NeQ-TEC from  March -
December during quiet period.




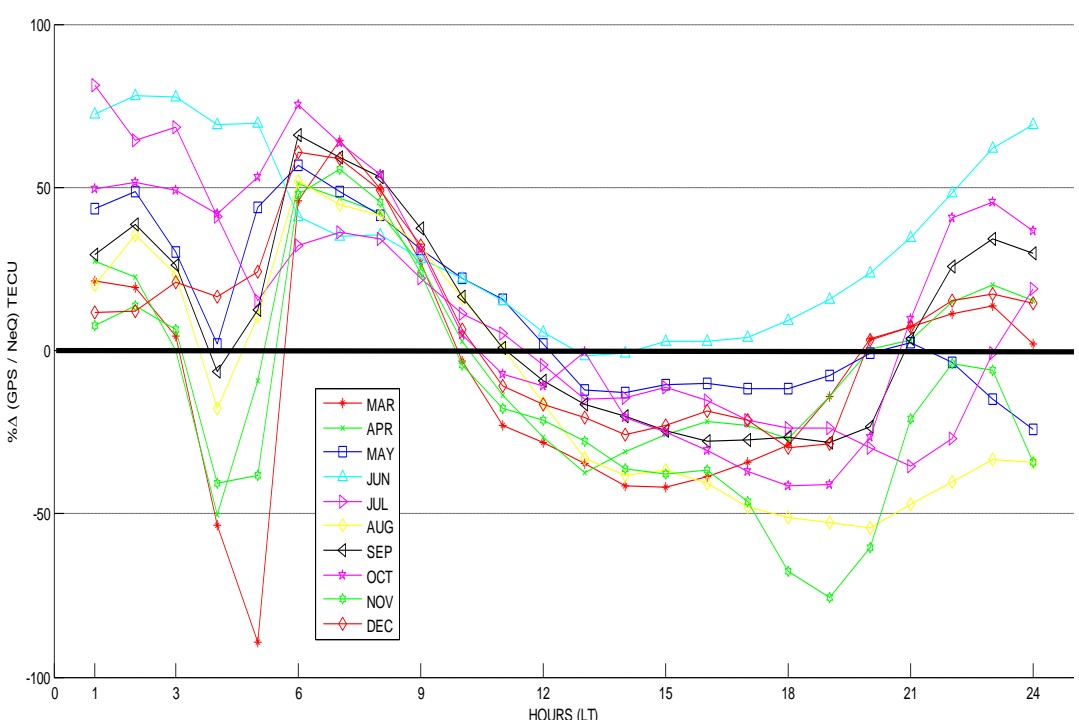

Figure 4b. The mass plot of the hourly % Δ TEC variations between the GPS-TEC and NeQ-
TEC from March - December during quiet period




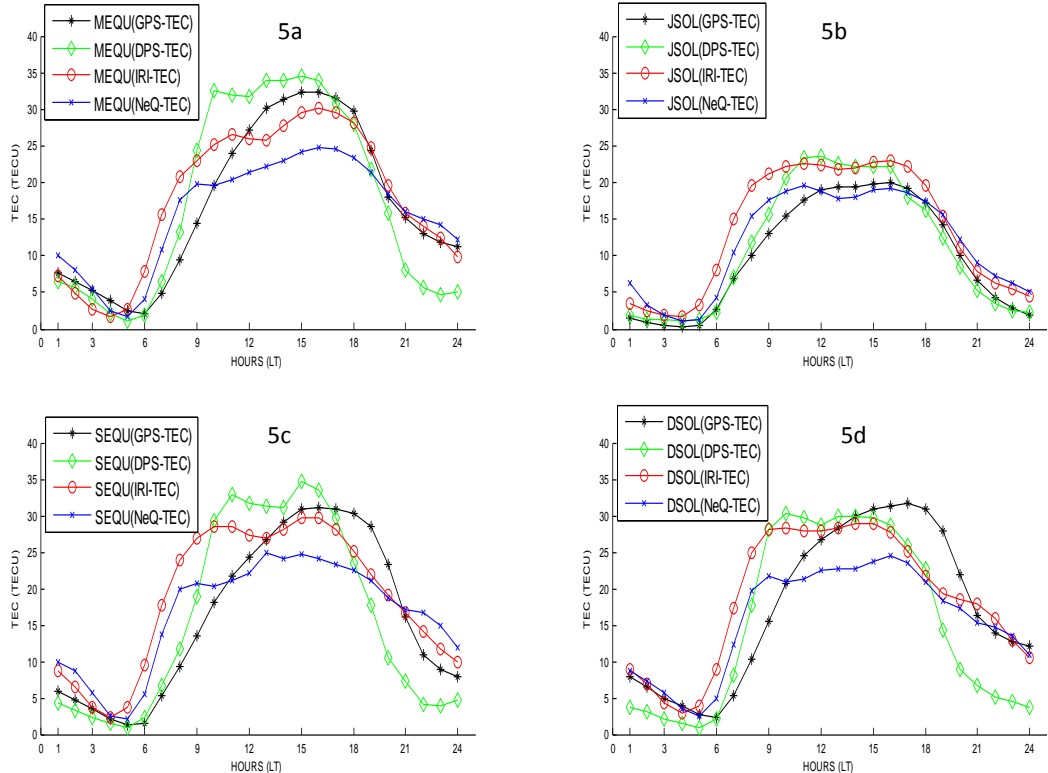

Figure 5a. The hourly variations of median GPS-TEC, DPS-TEC, IRI-TEC, and NeQ-TEC for March equinox during quiet period.

Figure 5b. The hourly variations of median GPS-TEC, DPS-TEC, IRI-TEC, and NeQ-TEC for June solstice during quiet period.

Figure 5c. The hourly variations of median GPS-TEC, DPS-TEC, IRI-TEC, and NeQ-TEC for September equinox during quiet period.

Figure 5c. The hourly variations of median GPS-TEC, DPS-TEC, IRI-TEC, and NeQ-TEC for December solstice during quiet period.