# Peer review of "Morphology of GPS and DPS-TEC over an equatorial station: validation of IRI and NeQuick 2 models."

_Annales Geophysicae, 2018_

## Referee Comment (RC1) · Anonymous Referee #1 · 16 Jul 2018

The manuscript show an interesting work on the comparison between GPS-TEC and DPS-TEC, and on the validation of IRI-TEC and NeQ-TEC, at equatorial latitude, and during low solar activity. The authors found that there is a solar zenith angle dependence of the variations in GPS-TEC DPS-TEC, IRI-TEC and NeQ-TEC. They also show a faster increase of DPS-TEC, IRI-TEC and NeQ-TEC, with respect to GPS-TEC, during sunrise. The authors suggest a misinterpretation of the topside Ne profile of the DPS-TEC, IRI-TEC and NeQ-TEC, due to the incorporation of the plasmaspheric electron content (PEC) into the models. Their conclusion is that the DPS-TEC is suitable to predict GPS-TEC during daytime when PEC contribution is often negligible, while should be paid attention when considering dusk period: a substantial correction is

needed. The paper describe an interesting investigation in order to improve models for the equatorial latitudes. I suggest the paper for final publication after some minor revisions.

Line 15: A better description of the geomagnetic conditions of the "quiet days " taken into account for the analysis, For example showing some geomagnetic index related to the considered periods

Line 176: there is a typing error in the equation Line 179: explain better what do you mean for "most quiet slant GPS-TEC data" Line 204 -205: delete "(Universal time)"

Line 211 – 213: The meaning of the sentence has to be better explained Line 230 : "GPSTEC" has to be replaced by "GPS-TEC" Line 285: "DPS-TEC constantly" has to be replaced by "DPS-TEC is constantly" Line 352: "slowly reached" has to be replaced by "slowly reaches" Line 353: "later decay" has to be replaced by "later decays" Line 437 – 439: The meaning of the sentence has to be better explained.

---

## Author Comment (AC1) · 17 Jul 2018

We thank the referee for their constructive comments and suggestions. We are very glad to alter many of the suggested changes. Please, find below our point-by-point replies to your suggestions.

Line 15: A better description of the geomagnetic conditions of the "quiet days " taken into account for the analysis, For example showing some geomagnetic index related to the considered periods.

The statement in Line 15 has been rewritten to accommodate geomagnetic in-

dex of the quiet days used: The five most quietest days with Ap≤4 of each month were employed for the investigation. The quietest days for the investigation were taken from the international quiet days (IQD) from the website http://www.ga.gov.au/oracle/geomag/iqd_form.jsp.

Line 176: there is a typing error in the equation

The typing error in Line 176 has been rewritten as: $S(E)=1/(cosâĄą(z))=[1-((R_E×cosâĄą(E))/(R_E+h_s ))^2 ]^{(-1/2)}$

Line 179: explain better what do you mean for "most quiet slant GPS-TEC data"

The explanation of most quiet slant GPS-TEC data is given as: The five most quiet slant GPS-TEC data are slant GPS-TEC data recorded from the GPS receiver during the five most quiet days from the international quiet days (IQD) from the website http://www.ga.gov.au/oracle/geomag/iqd_form.jsp of each month in the year 2010. The slant TEC of most quiet days is converted to vertical TEC of most quiet days with the expressions below.

ãĂŰ(GPS-TEC)ãĂŮ_(V    )=ãĂŰ(GPS-TEC)ãĂŮ_(S    )-[b_S+b_R+b_SR    ]/S(E)    1
$S(E)=1/(cosâĄą(z))=[1-((R_E×cosâĄą(E))/(R_E+h_s ))^2 ]^{(-1/2)}$ 2

Line 204 -205: delete "(Universal time)" The Universal time has been deleted and the statement now reads: Thus, 0100 UT is the same as 0200 LT in Nigeria.

Line 211 – 213: The meaning of the sentence has to be better explained.

The statement has been rewritten as: The median of the five most quietest days of each month was deduced. Therefore, the average of the median of the five most quietest days under a particular season discussed above was inferred to give GPS-TEC, DPS-TEC, IRI_TEC and NeQ-TEC a particular season.

Line 230: "GPSTEC" has to be replaced by "GPS-TEC" The statement now reads: $\Delta_{(GPS/DPS)}$, represents the change between GPS-TEC and DPS-TEC

Line 285: "DPS-TEC constantly" has to be replaced by "DPS-TEC is constantly" The word has been rewritten as DPS-TEC is constantly lower than the GPS-TEC

Line 352: "slowly reached" has to be replaced by "slowly reaches" The phrase has been changed to: slowly reaches

Line 353: "later decay" has to be replaced by "later decays" The phrase later decay has been changed to: later decays

Line 437 – 439: The meaning of the sentence has to be better explained. The statement has been rewritten as:

Our findings show that the variations in GPS, DPS, IRI, and NeQ-TEC are maximum and minimum around noontime and pre-sunrise or sunrise minimum indicating that both observed and modeled TEC are solar zenith angle dependence.

---

## Referee Comment (RC2) · Anonymous Referee #2 · 13 Aug 2018

The manuscript deals with the comparison of observed and modeled TEC over the African equatorial sector. Even though comparative studies of modeled outputs with observations from different sectors have importance in terms of modelling, the present paper is not well organized and presentation is monotonous with poor English. The manuscript needs substantial revisions to become suitable for publication. Some of the comments are outlined here under.

The English is at very primary level with several misleading, confusing and incomplete sentences. The manuscript requires a thorough English revision by a person well known with English

[Figure]

The introduction is too lengthy which must be reduced

The conclusion section must be rewritten outline main and new findings of the present study.

Line 176: …. Krishna software.. - Give reference and/or the place where it can be archived.

Line 190: Instead of giving infinity, the upper height limit of the ionosonde topside profile must be given in the integral limits of second part.

Lines 200-201: refine the sentence about relation between UT and LT

Lines 229-332: the sentences are too monotonous. Must be rewritten

Figure 5 title must be rewritten

Lines 369-370: It is not the interaction of electric and magnetic fields. It is because of the vertical drift caused due to the combined effect of mutually perpendicular electric and magnetic fields on the plasma.

The study is nominal comparison of TEC from different methods and model. Whereas the statement in lines 456-457 "This will reshape the model parameters for improved ionospheric modeling over Africa" is superstitious.

Figure 3b: Cross check the huge negative values in March or Dec.

I am attaching annotated manuscript with more corrections and suggestions.

Please also note the supplement to this comment:
https://www.ann-geophys-discuss.net/angeo-2018-57/angeo-2018-57-RC2-supplement.pdf

**Supplement:**

O.O. Odeyemi[1] , J.O Adeniyi[2] , O.A. Oladipo[3], A.O. Olawepo[3], I.A. Adimula[3], E.O 
[revised manuscript text omitted]

---

## Author Comment (AC2) · 29 Aug 2018

We thank the Referee 2 for the thoughtful and helpful comments. The changes have been effected accordingly, and highlighted in red color. Please, find also the point by point response to the change made to your comment in the body of the manuscript.

(A)Line 176:.... Krishna software.. - Give reference and/or the place where it can be archived.

The place where it can be achieved has been suggested below. (Global Positioning System total electron content analysis application user's manual, 2009, institute for

[Figure]

Scientific Research, Boston College, Chestnut Hill, Massachusetts)

(B)Line 190: Instead of giving infinity, the upper height limit of the ionosonde topside profile must be given in the integral limits of second part.

The upper height limit of the ionosode topside has been rewritten as:

TEC=$\int \_0^{hmF2} Ne\_B(dh) + \int \_{hmF2}^{1000} Ne\_T(dh)$3

(C)Lines 200-201: refine the sentence about relation between UT and LT

The statement in Line 200-201 has been rewritten to show clearly the relationship between UT and LT The universal time (UT) is the time standard for the record of GPS and DPS data but we converted UT to local time (LT) by adding one hour to corresponding UT. Nigeria is 1 hour in advance of Greenwich mean time (GMT) thus, 0100 UT is the same as 0200 LT in Ilorin, Nigeria.

(D)Lines 229-332: the sentences are too monotonous. Must be rewritten

The sentences in Lines 229 - 332 have been written for $\Delta\_(GPS/DPS)$, $\Delta\_(GPS/IRI)$, and $\Delta\_(GPS/NeQ)$, represent the changed TEC between GPS-TEC and DPS-TEC, GPS-TEC and IRI-TEC, and GPS-TEC and NeQ-TEC, respectively while %($\Delta\_(GPS/DPS)$ ), %($\Delta\_(GPS/IRI)$ ), and %($\Delta\_(GPS/NeQ)$ ), represent the percentage deviation between GPS-TEC and DPS-TEC, GPS-TEC and IRI-TEC, and GPS-TEC and NeQ-TEC, respectively.

(E)Figure 5 title must be rewritten

The Figure 5 has been written as: Fig 1b Seasonal variations of GPS-TEC, DPS-TEC, IRI-TEC and NeQ-TEC (i) March Equinox (i) June Solstice (ii) September Equinox (ii) December Solstice in 2010 over Ilorin during quiet periods.

(F)Lines 369-370: It is not the interaction of electric and magnetic fields.

The statement in Lines 369-370 has been written as suggested by the Referee Thus,

the bite-out results from the vertical drift due to the combined effect of mutually perpendicular electric and magnetic fields on the plasma.

(G)The study is nominal comparison of TEC from different methods and model. Whereas the statement in lines 456-457 "This will reshape the model parameters for improved ionospheric modeling over Africa" is superstitious.

The statement on lines 456-457 has been rewritten to capture the parameters investigated. The investigation will improve the modeled TEC for better performance over African region.

(H)Figure 3b: Cross check the huge negative values in March or Dec. The huge negative values still fall on March after checking.

(I)I am attaching annotated manuscript with more corrections and suggestions.

All the attached corrections and suggestions in the annotated manuscript have been altered as suggested by the Referee. Please find the attached edited manuscript.

(J) The English suggestion.

We have improve the English significantly

(K)The introduction is too lengthy which must be reduced

The introduction the paper has been reduced

(L)The conclusion section must be rewritten outline main and new findings of the present study.

The conclusion has been rewritten as: (i)We have investigated and compared the variations of observed and modeled TEC over an equatorial station of Africa during just ascending phase cycle of low solar activity in the year 2010. Our findings show that both the observed and modeled TEC are solar zenith angle dependent. (ii)Our observation revealed faster sunrise increase in the modeled TEC relative to GPS-TEC

which suggest the misinterpretation of the topside Ne profile of the modeled TEC in or-der to incorporate the plasmaspheric electron content (PEC). (iii)We also found equal daytime TEC between observed TEC and modeled TEC suggesting that the model TEC could represent GPS-TEC in the absence of plasmaspheric TEC contribution. (iv)We attributed the inconspicuous bite-out in the GPS-TEC during the daytime to the quick refill of fountain effect by higher rate of plasma production at the magnetic equator around noontime. (v)The discrepancy between GPS-TEC and modeled TEC during the dusk period requires attention in particular around 0400 - 0500 LT that shows the highest percentage deviations. (vi) We also found that the overestimation of % ∆TECIRI-GPS in May and June at all hours of the day. (vii)Furthermore, the percentage deviations in DPS and modeled-TEC during dusk periods is always higher than their corresponding deviations during the daytime and the values of daytime deviation in DPS and NeQ-TEC are smaller compared to daytime deviation in IRI-TEC. This study was carried out during the quietest period of the year 2010; it will be of advantage to investigate and compare studies on the most disturbed days with our results. Moreover, additional stations around the equatorial region will be required to validate the latitudinal effect of the model TEC; this could improve the model parameters for better ionospheric modeling over African sector.

Please also note the supplement to this comment:
https://www.ann-geophys-discuss.net/angeo-2018-57/angeo-2018-57-AC2-supplement.pdf

**Supplement:**

[revised manuscript text omitted]

The TEC inferred from the measured GPS-, DPS- and modeled TEC have been estimated during quiet period in 2010 at the Ilorin station. Figure 1a shows the simultaneous plots of hourly variations of the monthly median of GPS-, DPS-, IRI-, and NeQ- TEC during quiet period. The GPS-TEC is plotted in black line with the star symbol; the DPS-TEC is in green line with the diamond symbol, IRI-TEC is in red line with zero symbols, and finally, the NeQ-TEC is in blue line with multiplication symbol. All TEC plots are regulated by the same local time (LT) on the horizontal axis. The result shows that the morphologies of GPS-, DPS-, modeled-TEC increase gradually from the sunrise period (0700 -0900 LT) and reach the daytime maximum, mostly around (1200 - 1700 LT), and then later decay steadily until minimum value is reached around 0600 LT. Therefore, our result suggest that the diurnal variations of the observed and modeled TEC capture the well known solar zenith angle dependence of TEC since they are all characterized with pre-sunrise minimum, daytime maximum, daytime depression (modeled TEC) and post-sunset decay. The lowest and highest pre-sunrise minimum were ranged from 0.66 TECU (DPS) - 4.49 TECU (DPS) while the lowest and highest daytime maximum were found between 17.75 TECU (NeQ) - 38.0 TECU (DPS). The noontime bite-out was observed in modeled TEC around 1200 LT and 1500 LT except in GPS-TEC where the bite-out was inconspicuous except that the shift in daytime maximum in GPS-TEC between 1500 and 1700 LT in all months. The two moderate peaks (pre-noon peak and post-noon peak) observed in DPS-TEC and modeled TEC were due to the effect of bite-out on the modeled TEC signature. However, the pre-noon and post-noon peaks were absent in the variation of GPS-TEC since the noontime bite-out is not well captured in GPS-TEC morphology. We also found that around the sunrise period, the model TEC rises faster than the GPS-TEC but IRI-TEC rises faster compared to DPS-TEC and IRI-TEC. Between 0600 and 0900 LT, the lowest and highest difference in the rises of IRI-TEC compared to GPS-TEC were ~ 5.0 TECU (March) and 15.3 TECU (November), respectively. The post noontime decay was faster in DPS-TEC compared to GPS-TEC and modeled TEC in all months.  Figure 1b reveals the simultaneous seasonal variations of GPS-; DPS-; and modeled-TEC during quiet period of (i) March Equinox, (ii) June solstice, (iii) September equinox and (iv) December solstice. The daytime maximum  ranges are between   ~ 24.8 TECU  (NeQ) - ~ 34 TECU  (DPS), ~ 19.2 TECU (NeQ) - ~ 22.6 TECU (DPS), ~ 24.9

TECU (NeQ) - ~ 33.5 TECU (DPS) and ~ 24.55 TECU (NeQ) - ~ 31 TECU (DPS), in March equinox, June solstice, September equinox, and December solstice, respectively.  We observed that GPS-TEC and modeled TEC were maximum and minimum TEC at March equinox and June solstice indicating semi-annual variation in TEC.

**3.2 Percentage deviation of DPS-TEC; IRI-TEC; and NeQ-TEC**

Figures 2a, 3a and 4a and Figures 2b, 3b, and 4b show hourly variations of ΔTEC and mass plot of hourly variations of % ΔTEC, respectively between GPS-TEC and DPS-; IRI-; and NeQ-TEC from March to December during quiet period**.** In Figure 2a and 2b, the overestimated ΔTEC$_{DPS-GPS}$ is found within ~ 5.13 TECU (March) - ~ 19.12 TECU (July) around 0700 - 1600 LT while the underestimated ΔTEC$_{DPS-GPS}$ is ranged between ~ 3.2 TECU (June) - ~ 16.4 TECU (November) around 1700 - 2400 LT. The overestimation and underestimation of %Δ$_{IRI-GPS}$ are ranged from ~ 2% - ~ 49% and ~ - 1.36% - ~ - 306%, respectively. From Figures 3a and 3b, the overestimated ΔTEC$_{IRI-GPS}$ occurred regularly around 0400 - 1200 LT in all months. The overestimated and underestimated ΔTEC$_{IRI-GPS}$ ranges between ~ 9.13 TECU (July) - ~ 15.3 TECU (November) and ~ 0.15 TECU (October) - ~ 0.95 TECU (July), respectively. However, a few underestimation and overestimation of ΔTEC $_{IRI-GPS}$ occur irregularly around 1300 - 0300 LT in all months.  We also found that IRI-TEC completely overestimated GPS-TEC in May and June between 0100 and 2400 LT. The overestimation of %ΔTEC $_{IRI-GPS}$ ranges between ~ 0.1% to ~ 86% in all months.  In Figures 4a and 4b, NeQ-TEC overestimated GPS-TEC within 0100 - 1100 LT and 2000 - 2400 LT with ΔTEC $_{NeQ-GPS}$ ranges from ~ 9.72 (September) and  ~ 0.01 (April). We also found that NeQ-TEC underestimated ΔTEC $_{NeQ-GPS}$ is between ~ 9.72 (Nov) - ~ 0.11(May). The overestimation and underestimation of %ΔTEC $_{NeQ-GPS}$ are within  ~ 0.02%  - ~ 81% and ~ - 0.3% - ~ - 75% respectively.

**3.3 Comparisons of the percentage deviations from GPS-TEC**

From Figure 2b, 3b, and 4b, The percentage deviation between GPS- and DPS-TEC are greater than 100% in March August, September, November and December between 0400 - 0500 LT and around 2200 - 2400 LT in June and July. The percentage deviation between GPS- and IRI-TEC are also lower than 100% except in March around 0400 LT whereas the deviation between GPS- and IRI-TEC is greater than 100%. The percentage deviations in DPS and modeled-TEC during

dusk periods are always higher than their corresponding deviations during the daytime. During the daytime, the values of deviation are small in DPS and NeQ-TEC compared to IRI-TEC.

**4.0 Discussion of Result**

An investigation into the variations of GPS-TEC, DPS-TEC, IRI-TEC, and NeQ-TEC at an equatorial station ($8.5^0$N $4.65^0$ E) in Africa during low solar activity in the year 2010 has been carried out. The TEC increases gradually from the sunrise period, then slowly reaches the daytime maximum, and later decays till the pre-sunrise minimum. This result indicates that the TEC is a solar zenith angle dependence indicating maximum and minimum TEC during the noontime and dusk time, respectively (Wu et al.2008; Aravindan and Iyer 1990; and Kumar and Singh 2009). Interestingly, our result that reveal the faster rise in the DPS-TEC compared to GPS-TEC during sunrise is not consistent with the findings of Ezquer et al. (1992) at Tucumán (26.9° S; 65.4° W), Belehaki et al. (2004) at Athens, McNamara (1985) at low latitude and Obrou et al. (2008) at Korhogo (9.33°N, 5.43°W, Dip = 0.67°S). They all found that the variation of GPS-TEC rises faster than the DPS-TEC during the sunrise. The evidence of PEC on GPS-TEC was recently reported by Belehaki et al. (2004). They extracted the plasmaspheric electron content (PEC) from the GPS-TEC and found a significant PEC in the morning and evening. Also, Jodogne et al. (2004), Mosert et al. (2007), and Mckinnell et al. (2007) obtained a rough estimation of PEC from the GPS and DPS-TEC. They concluded that the combined GPS-TEC and DPS-TEC could give the PEC of a given location. Therefore, the higher rise in DPS-TEC compared to GPS-TEC during the sunrise could be attributed to inaccurate representation of PEC in the topside DPS-TEC profile during the extrapolation from the peak of NmF2 to around ~ 1000 km of Ne profile. Since, a typical TEC measurement naturally includes the PEC contributions (Belehaki et al. 2003; Balan and Iyer, 1983; Carlson, 1996; and Breed et al, 1997). The higher values in DPS-TEC compared with IRI-TEC around sunrise is not consistent with Rios et al. (2007) who investigated the comparison of DPS-TEC and IRI-TEC. They found that DPS-TEC is smaller than IRI TEC in all hours. They concluded that the prediction of IRI-TEC included the high topside Ne profile. Thus, our observation suggests that the IRI-TEC has incorporated low topside Ne profile in the IRI model or the excessive exaggeration of PEC contribution in the topside Ne profile in the DPS-TEC.

Our finding that reveals  equal  daytime GPS-TEC and DPS-TEC in April, August and December may suggests that the topside Ne profile in GPS-TEC and is well captured in the DPS-TEC topside profile the absence of or negligible PEC in DPS-TEC values. The insignificance of daytime PEC has been reported by Rastogi et al. (1971) and  Belehaki et al. (2004). Our higher daytime DPS-TEC compared with  daytime IRI-TEC is consistent with McNamara (1985). However, Obrou et al. (2008) at the equatorial station, found higher the IRI-TEC than the DPS-TEC at the low solar activity.  Therefore, the reduced daytime IRI-TEC compared with GPS-TEC indicates the excessive PEC removal from the model values that its PEC contribution had been initially exaggerated during the sunrise.  Furthermore, the reduced NeQ-TEC values compared to GPS-TEC in all months is consistent with the report of  Migoya-Orue et al. (2017), Zakharenkova (2016), Rabiu et al., (2014) and  Nava and Radicella  (2009). They concluded that the PEC contribution on topside NeQ profile is required for the accurate prediction of the model.

The daytime bite-out in TEC is linked to the occurrence of the most active fountain effect during the noontime at the magnetic equator. The bite-out results from the vertical drift which is the combined effect of mutually perpendicular electric and magnetic fields on the plasma.  The drift lifts the plasma at the magnetic equator and diffused along geomagnetic field lines into the high latitudes, therefore, leaving the reduced TEC at the magnetic equator (Bandyopadhyay, 1970; Olwendo et al., 2013; Skinner, 1966; Bolaji et al., 2012). However, the absence of daytime bite-out (Olatunji, 1967) in GPS-TEC in our finding may be due to the greater productions of the bottomside and topside electron content that are enhanced quickly to replenish the loss of the ionization that occurs through the fountain effect during the noontime.

The hourly variations of percentage difference between GPS-TEC and all models TEC reveal that the pre-sunrise values in DPS-TEC, IRI-TEC and NeQ-TEC require attention due to high percentage difference recorded in all variations especially in March for DPS-TEC and the models, and November and December for DPS-TEC only.  The daytime DPS-TEC value is closer to the GPS-TEC value compared to the daytime IRI-TEC and NeQ-TEC values. The nighttime NeQ-TEC and IRI-TEC perform better with GPS-TEC compared with DPS-TEC in all

months. However, more improvement is also required to minimize the effect of the discrepancies observed during the dusk periods. More work need to be done during the pre-sunrise in all models especially in March for all models, and November and December for DPS-TEC.

Seasonally, we found that TEC is maximum and minimum during the equinoxes and the solstices, respectively. Our report is consistent with Mala et al. (2009), Wu et al. (2008), Kumar and Singh (2009), and Balan and Rao, (1984) who investigated TEC in various regions. They concluded that the seasonal variation in TEC is attributed to the seasonal differences in thermospheric composition. Moreover, the sub-solar point is around the equator during the equinox. Consequently, the sun shines directly over the equatorial region, and in addition to the high ratio of $O/N_2$ around the region, this translates to stronger ionization, thus, semi-annual pattern is formed. Our finding is supported by Ross Skinner, (1966), Bolaji et al. (2012), and Scherliess and Fejer (1999) who obtained semi-annual variation in TEC. Scherliess and Fejer (1999) also concluded that daytime $E \times B$ drift velocity could result to semi-annual variation because the drift is more and less significant in the equinoctial months and June solstice, respectively.

**5.0 Conclusion**

(i)We have investigated and compared the variations of observed and modeled TEC over an equatorial station of Africa during just ascending phase cycle of low solar activity in the year 2010. Our findings show that both the observed and modeled TEC are solar zenith angle dependent.

(ii)We reported the inconspicuous of the noontime bite-out in the observed TEC (Olatunji, 1967) which may be due to the quick refill of fountain effect by higher rate of plasma production at the magnetic equator around noontime.

(iii)Furthermore, our observation revealed faster sunrise increase in the modeled TEC relative to GPS-TEC which suggest the misinterpretation of the topside Ne profile of the modeled TEC in order to incorporate the plasmaspheric electron content (PEC).

(iv)We found equal daytime TEC between observed TEC and modeled TEC suggesting that the model TEC could represent GPS-TEC in the absence of plasmaspheric TEC contribution.

(v)The discrepancy between GPS-TEC and modeled TEC during the dusk period requires attention in particular around 0400 - 0500 LT that shows the highest percentage deviations.

(vi) We also found that the overestimation $\Delta$TEC of IRI-TEC in May and June in all hours of the day.

(vii)We found that the percentage of deviations in DPS and modeled-TEC during dusk periods is always higher than their corresponding deviations during the daytime and the values of deviation are small in DPS and NeQ-TEC compared to IRI-TEC during the daytime.

This study was carried out during the quietest period of the year 2010; it will be of advantage to investigate and compare studies on the most disturbed days with our results. Moreover, additional stations around the equatorial region will be required to validate the latitudinal effect of the model TEC; this could improve the model parameters for better ionospheric modeling over African sector.

**6.0 Acknowledgments**

We would like to appreciate Massachusetts University and Boston College, respectively for the donations of DPS and GPS facilities to the University of Ilorin, Nigeria. Also, the kind support of the University of Ilorin in running the DPS and GPS stations and in providing the data for use are gratefully acknowledged and appreciated.

---

## Author Response (AR1)

We thank the Referee 1 for their constructive comments and suggestions. We are very glad to alter many of the suggested changes. Please, find below our point-by-point replies to your suggestions.

Referee1 comment
(A)Line 15: A better description of the geomagnetic conditions of the "quiet days " taken into account for the analysis, For example showing some geomagnetic index related to the considered periods.

Authors Response
The statement in the Line 15 has been rewritten to accommodate geomagnetic index of the quiet days used:
The five quietest days with Ap ≤ 4 of each month were employed for the investigation. The quietest days for the investigation were taken from the international quiet days (IQD) from the website http://www.ga.gov.au/oracle/geomag/iqd_form.jsp. However, the Reviewer 2 suggested that "the statement is not necessary in the abstract". Thus, it has been deleted in the section.

Referee1 comment
(B)Line 176: there is a typing error in the equation

Authors Response
The typing error in Line 176 has been rewritten as:
$$S(E) = \frac{1}{\cos(z)} = \left[1 - \left(\frac{R_E \times \cos(E)}{R_E + h_s}\right)^2\right]^{-1/2}$$

Referee1 comment
(C)Line 179: explain better what do you mean for "most quiet slant GPS-TEC data"

Authors Response
The explanation of most quiet slant GPS-TEC data is given as:
Most quiet slant GPS-TEC data are the quietest slant GPS-TEC data acquired by GPS receiver when the geomagnetic variations are a minimum in each month, and obtained from the international quiet days (IQDs) (http://www.ga.gov.au/oracle/geomag/iqd_form.jsp) in the year 2010. The slant TEC of quietest days is converted to vertical TEC of quietest days with the expressions below.

$$(GPS - TEC)_V = (GPS - TEC)_S - [b_S + b_R + b_{SR}]/S(E) \qquad 1$$
$$S(E) = \frac{1}{\cos(z)} = \left[1 - \left(\frac{R_E \times \cos(E)}{R_E + h_s}\right)^2\right]^{-1/2} \qquad 2$$

Referee1 comment
(D)Line 204 -205: delete "(Universal time)"

Authors Response
The Universal time has been deleted and the statement now reads:
The universal time (UT) is the time standard for the record of GPS and DPS data, but we converted UT to local time (LT) by adding one hour to corresponding UT. Nigeria is 1 hour in

advance of Greenwich Mean Time (GMT) thus, 0100 UT is the same as 0200 LT in Ilorin, Nigeria.

Referee1 comment
(E)Line 211 – 213: The meaning of the sentence has to be better explained.

Authors Response
The statement has been rewritten as:

The monthly median of the five quietest days were deduced and the average of the monthly median under a particular season as defined above to infer seasonal variations under GPS-TEC, DPS-TEC, IRI-TEC, and NeQ-TEC.

Referee1 comment
(F)Line 230 : "GPSTEC" has to be replaced by "GPS-TEC"

Authors Response
 The statement now reads:
$\Delta_{DPS-GPS}$ , represents the difference between GPS-TEC and DPS-TEC

Referee1 comment
(G)Line 285: "DPS-TEC constantly" has to be replaced by "DPS-TEC is constantly"

Authors Response
The word has been rewritten as
DPS-TEC is constantly lower than the GPS-TEC.
The phrase has been restructured due to correction made by Reviewer 2

Referee1 comment
(H)Line 352: "slowly reached" has to be replaced by "slowly reaches"

Authors Response
The phrase has been changed to:
slowly reaches

Referee1 comment
(I)Line 353: "later decay" has to be replaced by "later decays"

Authors Response
The phrase later decay has been changed to:
later decays

Referee1 comment
(J)Line 437 – 439: The meaning of the sentence has to be better explained.

Authors Response
The statement has been rewritten as:

The TEC increases gradually from the sunrise period, then slowly reaches the daytime maximum, and later decays to the pre-sunrise minimum. This result indicates that the observed and modeled-TEC are a solar zenith angle dependence showing peak and least TEC values during the noontime and dusk time, respectively.

We thank the Referee 2 for the thoughtful and helpful comments. The changes have been effected accordingly, and highlighted in red color. Please, find also the point by point response to the change made to your comment in the body of the manuscript. We also highlighted the changes in the body of the manuscript with red color.

Referee 2 comment
(A)Line 176:.... Krishna software.. - Give reference and/or the place where it can be archived.

Response
The place where it can be achieved has been suggested below.
(Global Positioning System total electron content analysis application user's manual, 2009, institute for Scientific Research, Boston College, Chestnut Hill, Massachusetts)

Referee 2 comment
(B)Line 190: Instead of giving infinity, the upper height limit of the ionosonde topside profile must be given in the integral limits of second part.

Response
The upper height limit of the ionosode topside has been rewritten as:

$$\text{TEC} = \int_0^{\text{hmF 2}} \text{Ne}_B(\text{dh}) + \int_{\text{hmF 2}}^{1000} \text{Ne}_T\,(\text{dh}) \qquad\qquad 3$$

Referee 2 comment
(C)Lines 200-201: refine the sentence about relation between UT and LT

Response
The statement in Line 200-201 has been rewritten to show clearly the relationship between UT and LT
The universal time (UT) is the time standard for the record of GPS and DPS data but we converted UT to local time (LT) by adding one hour to corresponding UT. Nigeria is 1 hour in advance of Greenwich mean time (GMT) thus, 0100 UT is the same as 0200 LT in Ilorin, Nigeria.

Referee 2 comment
(D)Lines 229-332: the sentences are too monotonous. Must be rewritten

Response
The sentences in Lines 229 - 332 have been written for
$\Delta_{\text{DPS}-\text{GPS}}$, $\Delta_{\text{IRI}-\text{GPS}}$, and $\Delta_{\text{NeQ}-\text{GPS}}$, represent the changed TEC between GPS-TEC and DPS-TEC, GPS-TEC and IRI-TEC, and GPS-TEC and NeQ-TEC, respectively while $\%(\Delta_{\text{DPS}-\text{GPS}})$,

$\%(\Delta_{IRI-GPS})$, and $\%(\Delta_{NeQ-GPS})$, represent the percentage deviation between GPS-TEC and DPS-TEC, GPS-TEC and IRI-TEC, and GPS-TEC and NeQ-TEC, respectively.

Referee 2 comment
(E)Figure 5 title must be rewritten

Response
The Figure 5 title has been written as:
Fig. 1b Seasonal variations of GPS-TEC, DPS-TEC, IRI-TEC and NeQ-TEC for: (i)March Equinox, (ii)June Solstice, (iii)September Equinox, and (iv)December Solstice over Ilorin during quiet periods in 2010. The line colors and symbols are the same as for diurnal variation in Figure 1a for all seasons.

Referee 2 comment
(F)Lines 369-370: It is not the interaction of electric and magnetic fields.

Response
The statement in Lines 369-370 has been written as suggested by the Referee
The bite-out results from the vertical plasma drift due to the combined consequence of mutually perpendicular electric and magnetic fields on the plasma

Referee 2 comment
(G)The study is nominal comparison of TEC from different methods and model. Whereas the statement in lines 456-457 "This will reshape the model parameters for improved ionospheric modeling over Africa" is superstitious.

Response
The statement on lines 456-457 has been deleted

Referee 2 comment
(H)Figure 3b: Cross check the huge negative values in March or Dec.

Response
I have checked and found that the huge negative values is still on March.

Referee 2 comment
(I)I am attaching annotated manuscript with more corrections and suggestions.

Response
All the attached corrections and suggestions in the annotated manuscript have been altered as suggested by the Referee. Please find the attached edited manuscript for your perusal.

Referee 2 comment
(J) The English suggestion.

Response
We have improved significantly the grammar and tense of the manuscript.

Referee 2 comment

(K)The introduction is too lengthy which must be reduced.

Response
The introduction of the paper has been reduced

[revised manuscript text omitted]

Referee 2 comment
(L)The conclusion section must be rewritten outline main and new findings of the present study.

Response
The conclusion has been rewritten as:

[revised manuscript text omitted]

---

## Author Response (AR2)

We will like to thank the editor and the reviewers for taking the time to correct our manuscript. We much appreciate their combined effort in making the paper suitable for publication. The technical corrections suggested by one of the reviewers have been amended and highlight in color red accordingly.

* Conclusions: remove '(i)' before the first sentence
The 'i' before the first sentence, in conclusion, has been deleted
We have examined the variations of observed and modeled TEC over an equatorial region in Africa during a year of low solar activity.

* Line 359: Correct 'Our findings showed:' as 'Our findings showed that:'
Line 359 has been written as suggested by the reviewer
Our findings showed that

* Line 360: remove 'that.'
Line 360 has been rewritten to remove that. The statement now reads
(i) GPS-TEC and modeled TEC are solar zenith angle dependence.